# Suppression of Chitin-Triggered Immunity by a New Fungal Chitin-Binding Effector Resulting from Alternative Splicing of a Chitin Deacetylase Gene

**DOI:** 10.3390/jof8101022

**Published:** 2022-09-28

**Authors:** Jesús M. Martínez-Cruz, Álvaro Polonio, Laura Ruiz-Jiménez, Alejandra Vielba-Fernández, Jesús Hierrezuelo, Diego Romero, Antonio de Vicente, Dolores Fernández-Ortuño, Alejandro Pérez-García

**Affiliations:** 1Departamento de Microbiología, Facultad de Ciencias, Universidad de Málaga, 29071 Malaga, Spain; 2Instituto de Hortofruticultura Subtropical y Mediterránea “La Mayora”, Universidad de Málaga, Consejo Superior de Investigaciones Científicas (IHSM-UMA-CSIC), 29071 Malaga, Spain

**Keywords:** cucurbits, *Podosphaera xanthii*, powdery mildews, chitin-binding effector, chitin deacetylase, chitin-triggered immunity

## Abstract

Phytopathogenic fungi have evolved mechanisms to manipulate plant defences, such as chitin-triggered immunity, a plant defensive response based on the recognition of chitin oligomers by plant-specific receptors. To cope with chitin resistance, fungal pathogens have developed different strategies to prevent chitin recognition, such as binding, breaking, or modifying immunogenic oligomers. In powdery mildew fungi, the activity of chitin deacetylase (CDA) is crucial for this purpose, since silencing of the *CDA* gene leads to a rapid activation of chitin signalling and the subsequent suppression of fungal growth. In this work, we have identified an unusually short *CDA* transcript in *Podosphaera xanthii*, the cucurbit powdery mildew pathogen. This transcript, designated *PxCDA3*, appears to encode a truncated version of CDA resulting from an alternative splicing of the *PxCDA* gene, which lacked most of the chitin deacetylase activity domain but retained the carbohydrate-binding module. Experiments with the recombinant protein showed its ability to bind to chitin oligomers and prevent the activation of chitin signalling. Furthermore, the use of fluorescent fusion proteins allowed its localization in plant papillae at pathogen penetration sites. Our results suggest the occurrence of a new fungal chitin-binding effector, designated CHBE, involved in the manipulation of chitin-triggered immunity in powdery mildew fungi.

## 1. Introduction

Powdery mildews (Erysiphales, Ascomycota) are obligate biotrophic fungi capable of causing diseases in a wide variety of monocotyledonous and dicotyledonous species across the world, including economically important crops such as cereals, grapevine, and a high number of vegetables and ornamental species [1,2]. Powdery mildew fungi are obligate biotrophic parasites, meaning that they need host living cells to grow and complete their life cycles. Biotrophy in these fungi is the result of developing haustoria inside plant cells, specialized structures of parasitism that are responsible for nutrient uptake and secretion of effectors [3,4]. Various vegetable crops are susceptible to powdery mildews, but cucurbits are arguably the group most affected. *Podosphaera xanthii* is the main causal agent of powdery mildew disease in cucurbits and one of the most important limiting factors for crop productivity worldwide [5]. To decipher the molecular basis of *P. xanthii* biology, our laboratory has invested considerable efforts in producing fundamental research resources such as the epiphytic and haustorial transcriptomes, RNA-seq data of early stages of the infection in melon, and the first draft genome [6,7,8,9]. In addition, we have developed specific research tools, such as transformation and RNAi silencing protocols [10,11,12,13]. All this has led to *P. xanthii* becoming probably the most versatile model species for powdery mildew research.

Due to their lifestyle as obligate biotrophs, powdery mildew fungi require the establishment of an intimate relationship with their hosts, with avoidance recognition by the plant being particularly critical. Plants have developed specific receptors to sense conserved microbe-specific molecules referred to as microbe- or pathogen-associated molecular patterns (MAPMs or PAMPs). Chitin is a major component of the fungal cell wall and one of the best studied fungal PAMPs. Chitin is a long-chain polymer of β-1,4-*N*-acetylglucosamine that provides structural rigidity to the fungal cell wall and acts as the first line of defence of phytopathogenic fungi against plant-secreted enzymes [14,15,16,17,18,19]. As a consequence of the activity of plant chitinases deployed in response to fungal attack, chitin oligomers are released from fungal cell walls that can be recognized by specific plant receptors, such as the chitin elicitor-binding protein (CEBiP) and the chitin elicitor receptor kinase (CERK1), activating the plant response termed chitin-triggered immunity [19,20]. Following chitin perception, plants produce defence mechanisms that include the accumulation of toxic compounds such as reactive oxygen species (ROS), strengthening of plant cell walls by the accumulation of cell wall deposits such as callose and lignin, and even plant cell death [20,21,22,23].

One of the most important conditions for the successful development of a fungal infection is to prevent the activation of PAMP-triggered immunity [24]. Therefore, a crucial aspect for fungal survival in plant environments is to hide cell wall PAMPs that remain unnoticed by the host. For this, the fungal effectors, small proteins secreted by fungal pathogens, play a key role, with phytopathogenic fungi having evolved different strategies to overcome chitin recognition. For example, in *Cladosporium fulvum*, two effectors have been identified with a role in altering chitin perception by the plant. AVR4 is an effector protein that binds to the chitin of the fungal cell wall, protecting against plant chitinases [21]. ECP6 is a chitin-binding effector containing LysM domains that sequesters the free immunogenic chitin oligomers released by the activity of plant chitinases, thus avoiding their recognition by the plant [16]. Homologues of ECP6 have been found in many fungi, and some of these homologues have been functionally analysed [20]. Effectors with similar chitin-binding functions have been described in other systems, such as the cacao pathogen *Moniliophthora perniciosa*, which secretes enzymatically inactive chitinases that retain substrate-binding specificity and prevent the activation of chitin-triggered immunity [25]. Some effectors act directly or indirectly on plant chitinases. For instance, the causal agent of septoria tritici blotch of wheat *Mycosphaerella graminicola* secretes the Mg3LysM effector, which interacts with chitinases either directly or indirectly through binding to chitinase-bound chitin molecules to prevent hydrolysis and the release of chitin fragments [26].

Another fungal strategy to suppress chitin signalling is the degradation of immunogenic oligomers. This degradation is the case for EWCA effectors, a family of secreted fungal chitinases that are released at pathogen penetration sites to break down immunogenic chitin fragments, thus preventing the activation of chitin-triggered immunity [27]. First identified in *P. xanthii*, EWCA proteins are widely distributed in pathogenic fungi. Remarkably, in the same fungus, a different enzyme with a similar function was found, the case for a haustorial-expressed lytic polysaccharide monooxygenase (LPMO), a new fungal LPMO that catalyses chitooligosaccharides, thus contributing to the suppression of plant immunity during haustorium development [28]. All this evidence shows that fungal pathogens have evolved to develop different strategies to solve a common problem to avoid the detection of chitin by the host plants.

Another mechanism involved in the battle for chitin recognition is chitin deacetylase (CDA). This protein is a conserved enzyme that catalyses the hydrolysis of the *N*-acetamido group in the *N*-acetylglucosamine units of chitin to convert it to chitosan, the deacetylated derivative of chitin, a poor substrate for plant chitinases and a compound with reduced elicitor activity compared to chitin [24,29]. CDA is a protein essential for virulence, and *cda* mutants of fungal pathogens such as *Verticillium dahliae*, *Fusarium oxysporum*, and *Ustilago maydis* showed remarkably reduced virulence [30,31]. Similarly, CDA is a key enzyme for *P. xanthii* development. RNAi silencing of the *PxCDA* gene resulted in a dramatic reduction in fungal growth that was linked to a rapid elicitation of chitin-triggered immunity [32]. In previous studies, two *P. xanthii* transcripts coding CDA enzymes differing in 30 nucleotides were found [6,32]. Recently, a third transcript also resulting from alternative splicing of the *PxCDA* gene was identified, seeming to encode a truncated version of a protein that lacks the chitin deacetylase active site but retains the chitin-binding domain. In this work, we examined the role of this putative chitin-binding effector in the suppression of chitin signalling. Our results demonstrated the chitin-binding properties of the protein expressed in vitro and its ability to prevent chitin recognition. In addition, localization studies showed that this protein is deployed in the plant papilla, exactly in the same location where chitin oligomers are abundantly present. Collectively, our results suggest that this new protein, designated chitin-binding effector (CHBE), is involved in the suppression of chitin-triggered immunity during fungal penetration. We anticipate that similar proteins are likely present in other fungal pathogens, at least in powdery mildew fungi.

## 2. Materials and Methods

### 2.1. Plants, Fungi, Bacteria, and Culture Conditions

Cotyledons of zucchini (*Cucurbita pepo* L.) cv. Negro Belleza (Semillas Fitó, Barcelona, Spain) and melon (*Cucumis melo* L.) cv. Rochet (Semillas Fitó) were used in this work. All plants were cultivated in a growth chamber at 24 °C under a 16 h light/8 h dark cycle. The *P. xanthii* isolate 2086 was grown on disinfected zucchini cotyledons that were maintained in 8 cm Petri dishes with Bertrand medium (sucrose 40 g L^−^^1^, agar 10 g L^−^^1^, benzimidazole 30 mg L^−^^1^, pH 7.0) at 22 °C under a 16 h light/8 h dark cycle for one week as previously described [33]. For fungal transformation, *Agrobacterium tumefaciens* strain C58C1 containing the proper plasmids was used. The strain C58C1 was grown at 28 °C in lysogeny broth (LB; “Lab-Lemco” powder 1 g L^−^^1^, yeast extract 2 g L^−^^1^, peptone 5 g L^−^^1^, sodium chloride 5 g L^−^^1^, pH 7.4) medium with rifampicin (50 µg mL^−1^) and tetracycline (5 µg mL^−1^) [11]. This medium was modified by adding the corresponding selective marker for each binary vector. For the maintenance, construction, and propagation of the different plasmids, the *Escherichia coli* strain DH5α was used. The resulting *E. coli* strains were grown at 37 °C in Luria-Bertani (LB) medium with the corresponding antibiotics [11].

### 2.2. Sequence Analysis and Protein Modelling

In previous studies, we identified two *P. xanthii CDA* transcripts (KX495502, KX495503) and one *CDA* gene [6,9,32]. In this work, a third transcript was found (KX495504). The number and localization introns in the *PxCDA* gene were analysed through the alignment of DNA and RNA sequences using MEGA 5 software (https://www.megasoftware.net/, accessed on 15 February 2022) [34]. The signal peptide was identified using the SignalP 4.1 server (https://services.healthtech.dtu.dk/service.php?SignalP-4.1, accessed on 15 February 2022) [35]. The search for conserve domains was carried out manually. Protein sequences were aligned using the UniProt web server (UniProt.org, accessed on 23 January 2022). The search for homologous amino acid sequences in databases was carried out using the BLAST tool (Basic Local Alignment Search Tool) from the National Center for Biotechnology Information (NCBI) (ncbi.nlm.nih.gov, accessed on 15 February 2022).

To predict the 3D structure of proteins, the AlphaFold (https://alphafold.ebi.ac.uk/) (accessed on 30 March 2022) and I-TASSER (http://zhanglab.ccmb.med.umich.edu/I-TASSER/) (accessed on 28 March 2022) web servers were used [36,37]. The 3D model of the *C. fulvum* ECP6 chitin-binding effector was obtained from the Research Collaboratory for Structural Bioinformatics (RCSB) protein data bank (http://www.rcsb.org/pdb/home/home.do, accessed on 28 March 2022).

### 2.3. RNA Extraction and cDNA Synthesis

RNA was isolated from *P. xanthii*-infected melon leaf tissue. For the inoculation, *P*. *xanthii* conidia were collected by immersing infected zucchini cotyledons in 50 mL of a 0.01% Tween-20/distilled water solution. Melon cotyledons were spray-inoculated with a spore suspension at 1 × 10^6^ conidia mL^−^^1^. Plants were then incubated in a growth chamber under the conditions mentioned above. For RNA isolation, infected leaf samples were finely ground in a mortar with liquid nitrogen and pestle. Subsequently, total RNA was isolated using the TRI Reagent RNA isolation system (Sigma-Aldrich, St. Louis, MO, USA) according to the manufacturer’s recommendation. Contaminating DNA was removed using a TURBO DNA-free kit (Invitrogen, Carlsbad, CA, USA). The total RNA concentration was estimated using a NanoDrop spectrophotometer ND-1000 (Thermo Fisher Scientific, Waltham, MA, USA). cDNA was synthesized using Superscript III Reverse Transcriptase (Invitrogen, Waltham, MA, USA) with Oligo dT(20) primers (Invitrogen) according to the manufacturer’s recommendation.

### 2.4. Plasmid Construction

All primers used for plasmid construction are listed in Appendix A. Polymerase chain reaction (PCR) products were amplified from cDNA obtained from *P. xanthii*-infected melon leaf samples taken 48 hpi (h postinoculation) using specific primers. For the construction of a protein expression vector, the specific primers CHBEexp-F/CHBE-STOP-R were used to amplify a fragment of 402 bp encoding PxCDA3, a potential chitin-binding effector (CHBE) derived from a *P. xanthii CDA* gene, to fuse it to a 6 × His tag. The amplified effector sequence excluded the signal peptide encoding sequence. To introduce the amplicons into the final vectors, we used Gateway cloning technology (Invitrogen). The *attB1* and *attB2* sequences were added to the N- and C-termini, respectively, of the amplicon by an additional PCR step using the primer pair attB1/attB2 [38]. Then, the PCR product was introduced into donor pDONR207 and destination pDEST17 vectors by BP and LR reactions as described by the manufacturer (Invitrogen).

For subcellular localization of CDA1 and CHBE (CDA3) proteins, plasmids containing translational fusions were used. To build the plasmids, the specific primers PxCDA-F/PxCDA1-noSTOP-R and PxCDA-F/PxCHBEnoSTOP-R were used to amplify fragments of 972 and 465 bp corresponding to the coding sequences of PxCDA1 and PxCHBE, respectively. For the green fluorescent protein (GFP) fusion (*PxCDA1-gfp*), the primers PgpdA-F/RPgpdA-PxCDA1 and Rgfp6-ApaI/Fgfp6-PxCDA1 were used to amplify the *PgpdA* promoter and *gfp* gene, respectively, using the backbone plasmid pPK2-hphgfp as a template. For the cyan fluorescent protein (CFP) fusion (*PxCHBE-cfp*), the primer pair Fcfp-CHBE/Rcfp-ApaI was used to amplify the *cfp* gene using the backbone plasmid pKM008 as a template. The three PCR fragments (promoter, fungal gene, and fluorescent marker gene) were joined using overlapping PCR and the primer pair PgpdA-F/Rgfp6-ApaI and PgpdA-F/Rcfp-ApaI for *PgpdA-PxCDA1-gfp* and *PgpdA-PxCHBE-cfp* fusions, respectively. The resulting fragments of 3.8 and 3.3 kb, respectively, were digested with *Kpn*I and *Apa*I restriction enzymes (New England Biolabs, Ipswich, MA, USA) and cloned into the plasmid pPK2-hphgfp [10], which was digested with the same enzyme, allowing the replacement of the original P*gpdA-hph-gfp* sequence with the P*gpdA-PxCDA1-gfp* and P*gpdA-CHBE1-cfp* sequences.

All the constructs were cloned and maintained in *E. coli* DH5α and were checked by enzymatic digestion and sequencing. The plasmid pPxCHBE-EXPRES was introduced into *E. coli* BL21-AI (Invitrogen) for the in vitro expression and purification of the CHBE protein. The plasmids pPxCDA1-GFP and pPxCHBE-CFP, containing the translational fusions *PxCDA1-gfp* and *pCHBE1-cfp* for subcellular localization studies, were introduced into *A. tumefaciens* C58C1 and used to transform *P. xanthii*.

### 2.5. Protein Expression and Purification

For in vitro expression of recombinant N-terminally 6-His tagged PxCDA3 protein, *E. coli* BL21-AI harbouring the pPxCHBE-EXPRES expression vector was used. For this purpose, *E. coli* cells were grown in LB medium supplemented with ampicillin (100 µg mL^−1^) at 37 °C and induced with 0.2% L-(+)-arabinose (Sigma-Aldrich, Steinheim, Germany) when they reached an OD_600nm_ of 0.4. Then, the cells were incubated overnight at 16 °C in an orbital shaker at 120 rpm. After incubation, the cells were collected by centrifugation at 8000× *g* for 8 min at 4 °C. The resulting pellet was lysed by adding 20 mL of washing buffer (50 mM sodium phosphate, 300 mM sodium chloride, 10 mM imidazole) supplemented with 1 mM phenylmethylsulfonyl fluoride (PMSF), 0.2 mg mL^−1^ lysozyme, and 1× cell lysis buffer (Sigma-Aldrich) with sonication pulses on ice. After unbroken cells and debris were removed by an additional centrifugation step, the soluble recombinant protein was purified using a HIS-Select Nickel Affinity Gel (Sigma-Aldrich) as described by the manufacturer. The purified recombinant proteins were dialyzed using a Lyzer Dialysis Cassette G2 10 K/15 mL (Thermo Fisher Scientific, Waltham, MA, USA). Protein purification was confirmed by sodium dodecyl sulfate-polyacrylamide gel electrophoresis (SDS-PAGE). Finally, the protein concentration was estimated by the Protein Concentration Calculator webserver (https://www.aatbio.com/tools/calculate-protein-concentration, accessed on 1 September 2022) using the absorbance value at 280 nm measured in an S-22 UV/Vis Spectrophotometer (BOECO, Hamburg, Germany), the extinction coefficient, and the molecular weight of the protein obtained from the Expasy webserver (http://www.expasy.org/, accessed on 1 September 2022).

### 2.6. Chitin-Binding Activity Assays

Prior to binding assays, colloidal chitin was prepared by partial hydrolysis of chitin with concentrated HCl [39]. Briefly, 20 mg of chitin from shrimp shells (Sigma-Aldrich) was manually ground in a mortar and pestle for 5 min. Then, the chitin powder was resuspended in 300 µL 37% HCl and incubated for 2 h at 4 °C with vigorous stirring. After incubation, to precipitate the chitin as colloidal chitin, 6 mL of distilled water was slowly added to the mixture. The suspension was collected by vacuum filtration through Whatman No. 4 filter paper (Sigma-Aldrich) and washed several times with distilled water until the pH of the suspension reached 7.0. Finally, the paste retained in the filter was resuspended in 2 mL of distilled water, obtaining a 1% (*w*/*v*) stock suspension of colloidal chitin that was stored at 4 °C in the dark until use.

To analyse the chitin-binding properties of His-tagged PxCDA3, binding assays were first carried out using colloidal chitin. In independent tubes, 250 μL of different substrate concentrations (5, 6, 8, and 10 mg mL^−1^) were prepared and centrifuged at 10,000× *g* for 5 min at 4 °C. The resulting pellets were washed in 250 μL of fresh 0.1 M phosphate-buffered saline (PBS), pH 7.0, and centrifuged again. The obtained pellets were resuspended in 250 μL of PBS buffer containing 100 μg mL^−1^ His-tagged PxCDA3 or 120 μg mL^−1^ bovine serum albumin (BSA) (negative control). After resuspending the pellets, the mixtures were incubated for 1 h on ice with stirring every 15 min. Later, the mixtures were centrifuged at 10,000× *g* for 5 min at 4 °C. After centrifugation, the supernatants and the pellets were separated. The proteins present in the supernatant samples were quantified as described above as an indicator of the proteins unbound to colloidal chitin.

The chitin-binding activity of His-tagged PxCDA3 was separately assayed using different chitin oligomers (GlcNAc)_1–7_ (IsoSep, Tullinge, Sweden). Prior to conducting the binding assays, a mix containing the seven chitin oligomers at a final concentration of 0.5 mg mL^−1^ was prepared. For this assay, His-tagged proteins were maintained bound to HIS-Select nickel affinity gel (Sigma-Aldrich) columns. After washing once with washing buffer, the proteins were incubated with a mix of chitin oligomers for 1 h at 4 °C with slight agitation. After incubation, the columns were washed once with washing buffer, and then the proteins were eluted with elution buffer. A portion of the reaction products was heated at 60 °C for 60 min to denature the proteins, filtered to eliminate the proteins, and directly analysed using a Jasco high performance liquid chromatography (HPLC) system equipped with a Luna 5 µm NH2 100 Å column (250 × 4.6 mm) (Phenomenex, Madrid, Spain). The mobile phase was 70% acetonitrile, and the flow rate was 0.7 mL min^−1^. The different chitin oligomers were detected by UV absorption at 210 nm. In parallel to the HPLC analysis, the reaction products were analysed by direct infusion in a Q Exactive Orbitrap mass spectrometer (MS) (Thermo Fisher Scientific) with electrospray ionization source H-ESI in positive mode at Ion Max source with a flow of 5 µL min^−1^ by 500 µL syringe (Hamilton, Reno, NV, USA).

### 2.7. Chitinase Activity and Chitin-Triggered Oxidative Burst Assay

The ability of PxCDA3 to protect fungal cell walls against hydrolysis by chitinases was studied using a chitinase activity assay against fungal growth. The assay was conducted in 96-well microtiter plates at a final volume of 100 µL. According to a previously described method [16], spore suspensions (1 × 10^4^ spore mL^−1^) of *Penicillium digitatum* were incubated in PDB (potato dextrose broth) medium overnight at 25 °C to promote spore germination, and *Botrytis cinerea* was incubated in CZAPEK medium overnight at 22 °C. Then, 40 µL aliquots of the liquid cultures were mixed with 60 µL of 120 µg mL^−1^ His-tagged PxCDA3 or *Streptomyces griseus* chitinases (Sigma-Aldrich) and deposited into 96-well microtiter plates. The mixtures were incubated for 24 h at 25 °C or 22 °C according to the temperature requirements of each fungus and then observed by light microscopy.

The ability of PxCDA3 to prevent chitin recognition was histochemically studied using a chitin-triggered oxidative burst assay. One-month-old melon cotyledons were infiltrated with different concentrations of colloidal chitin in the presence of different concentrations of His-tagged PxCDA3. The different mixtures of colloidal chitin/PxCDA3 were incubated for 1 h on ice. After incubation, leaves were infiltrated using a needleless syringe and maintained in a growth chamber with a 16 h light/8 h dark cycle at 24 °C. Twenty-four hours after infiltration, the chitin-triggered oxidative burst was histochemically analysed by the in situ accumulation of H_2_O_2_. For this analysis, H_2_O_2_ was detected using the 3,3-diaminobenzidine (DAB) method according to previously described methods [12].

### 2.8. Transformation by Growth onto Agroinfiltrated Tissues (TGAT) and Confocal Laser Scanning Microscopy (CLSM)

For subcellular localization of the PxCHBE (PxCDA3) and PxCDA1 proteins, the translational fusions PxCDA1-GFP and PxCHBE-CFP were introduced into *P. xanthii* by the TGAT transformation method, essentially as previously described [11]. Briefly, *A. tumefaciens* C58C1 cells harbouring the plasmids pPxCDA1-GFP and pPxCHBE-CFP were grown overnight at 28 °C in LB medium with the corresponding selection markers, induced with acetosyringone (Sigma-Aldrich) and co-infiltrated into melon cotyledons. Twenty-four hours after infiltration, cotyledons were treated with cefotaxime to kill *Agrobacterium* cells, and 24 h later, melon cotyledons were pulverized with a *P. xanthii* conidial suspension (1 × 10^5^ conidia mL^−1^) freshly prepared from infected zucchini cotyledons. Infected plants were maintained in a growth chamber with a 16 h light/8 h dark cycle at 24 °C until the development of transformants.

For subcellular localization of translational fusions, confocal laser scanning microscopy (CLSM) analysis was performed using a Leica SP5 II confocal microscope (Leica Microsystems, Wetzlar, Germany). The GFP and CFP fusion proteins were excited with 488 and 405 nm laser lines, and their fluorescence was detected at 495–530 and 460–485 bandpasses, respectively. For chitin and fungal membrane visualization, wheat germ agglutinin (WGA), Alexa Fluor 488 conjugate, and the membrane-specific stain FM4-64 were used as previously described [40]. Bright field images of the same areas were prepared using the transmission channel. All images were observed using a 63 × oil immersion objective and processed with the Leica LAS AF Lite 4.0 software (Leica Microsystems Inc., Wetzlar, Germany).

### 2.9. Real Time-Quantitative Polymerase Chain Reaction (RT-qPCR)

The expression analysis of fungal transcripts *PxCHBE* (*PxCDA3)*, *PxCDA1*, and *PxCDA2* was carried out by RT-qPCR. For this, cDNA was obtained from melon cotyledons infected with *P. xanthii* at 0, 24, 48, and 72 hpi as described above. The primers used for this analysis (Appendix A) were designed using Primer3 software (https://primer3.ut.ee/, accessed on 1 September 2022) [41]. The *P. xanthii* β-tubulin gene *PxTUB2* (KC333362) was used as a normalization reference gene [42]. RT-qPCR experiments were performed in a CFX384 Touch Real-Time PCR detection system (Bio-Rad Laboratories, Hercules, CA, USA) using SsoFast EvaGreen Supermix (Bio-Rad) according to the manufacturer’s instructions with the following cycling conditions: enzyme activation at 95 °C for 30 s, followed by 40 cycles at 95 °C for 5 s and 65 °C for 5 s. All reactions were performed in quadruplicate. After amplification, the data were analysed using Opticon Monitor analysis software version 2.02.24 (MJ Research, Waltham, MA, USA). Additionally, the amplicon sizes were confirmed by visualization on 2% agarose gels.

## 3. Results

### 3.1. Alternative Splicing of the PxCDA Gene Results in a Chitin-Binding Protein

In previous studies, two different *P. xanthii cytidine deaminase* (*CDA*) transcripts were identified, *PxCDA1* (KX495502) and *PxCDA2* (KX495503) [32]. In this work, a third transcript containing CDA sequences was found (KX495504). Appendix A and Figure 1A show the alignment of the cDNA sequences of *PxCDA* transcripts and the *PxCDA* gene (Appendix A) and the alignment of the corresponding mature proteins, that is, without a signal peptide (Figure 1A), respectively. A schematic representation of the *PxCDA1* gene, the transcripts, and their corresponding proteins are shown in Figure 1B. The transcript *PxCDA1* (whole CDA) is composed of 975 nucleotides and encodes a polypeptide of 325 amino acids. The transcript *PxCDA2* is slightly shorter; it has a size of 945 nucleotides and encodes a polypeptide of 314 amino acids due to an alternative splicing event that removes exon number 4. The third transcript, *PxCDA3*, is similar in size, 953 nucleotides. However, this transcript lacks 22 nucleotides in exon number 3, resulting in a premature STOP codon in exon number 5 and producing a 156-amino-acid polypeptide, half the size of the PxCDA1 and PxCDA2 proteins. Analysing the amino acid sequences of the mature proteins, PxCDA1 and PxCDA2 contain the carbohydrate-binding module at the N-terminus and the chitin deacetylase domain that encompasses much of the rest of the protein. PxCDA3, for its part, maintains the carbohydrate-binding module but loses much of the chitin deacetylase domain, presenting characteristics of a cysteine-rich protein, that is, a small size (156 residues) and an even number of cysteine residues (ten in total), nine of which are conserved in PxCDA1 and PxCDA2.

Since this truncated protein conserved the carbohydrate-binding module intact, we tested the ability of this protein to bind chitin. For this purpose, we expressed the protein in *E. coli* BL21-AI (Figure 2A). The protein was predominantly insoluble after lysis at room temperature, producing a large white pellet after centrifugation. However, lysis at 4 °C resulted in an acceptable increase in the protein in the soluble fraction. Purification of soluble proteins was conducted by immobilization on nickel affinity columns, obtaining His-tagged PxCDA3 with a yield of 112 µg mL^−1^ soluble protein. The chitin-binding ability of the PxCDA3 protein was studied via a binding assay using colloidal chitin as a substrate and BSA as a negative control. As shown in Figure 2B, the protein showed affinity for chitin, since the concentration of soluble protein decreased when increasing amounts of colloidal chitin were tested. In contrast, BSA was fully recovered from the supernatant, indicating the absence of binding.

### 3.2. PxCDA3 Protein Prevents Chitin Recognition by Sequestering Immunogenic Oligomers

To elucidate the physiological function of the PxCDA3 protein, we planned experiments to determine whether PxCDA3 had a role as a chitin-binding lectin such as *C. fulvum* AVR4 and the ability to protect fungal cell walls against hydrolysis by plant chitinases [21]. Alternatively, we also tested the putative role of PxCDA3 as a chitin-binding effector, such as *C. fulvum* ECP6, and the capability of the protein to sequester immunogenic chitin oligomers and to prevent chitin-triggered immunity [16]. Figure 3A shows the inability of His-tagged PxCDA3 to protect the hyphae of the fungi *P. digitatum* and *B. cinerea* against hydrolysis by *Streptomyces* chitinases, ruling out a role in preventing the hydrolysis of fungal cell walls by plant chitinases. In contrast, as shown in Figure 3B, PxCDA3 prevents chitin recognition by the plant, since the elicitor activity of colloidal chitin in terms of ROS production exemplified by H_2_O_2_ accumulation (oxidative burst) in melon leaves is considerably reduced in the presence of the in vitro expressed protein. This host immune suppression effect is quantified in Figure 3C, which shows the prevention of colloidal chitin-induced hydrogen peroxide accumulation by His-tagged PxCDA3 in a dose-dependent manner. These results suggested that the role of PxCDA3 in the *P. xanthii*–melon interaction could be comparable to the role of ECP6.

Chitin-triggered immunity in plants is activated after the detection of relatively large chitin oligomers with ≥6 units [24]. Therefore, one of the questions that arose after identifying PxCDA3-sequestered chitin oligosaccharides was what size of chitin oligomers were able to bind to this protein. After conducting a binding assay using a mix of chitin oligomers with different sizes (from 1 to 7 units), HPLC analysis showed that PxCDA3 bound only to oligomers equal to or greater than 5 units (Figure 4A) and showed a greater preference for chitin hexamers than pentamers, as revealed by the concentration of chitin oligomers detected by HPLC (Table 1). However, we noticed that the largest chitin oligomer assayed (heptamer) was more difficult to detect by HPLC. Therefore, we decided to analyse the samples by H-electrospray ionization (ESI)-MS. These analyses corroborated the presence of chitin pentamers and hexamers in the samples as previously observed by HPLC, but they also detected the presence of chitin heptamers (Figure 4B). Based on the ability of PxCDA3 to bind immunogenic chitin oligomers (hexamers and heptamers) to prevent elicitation of host immunity, we designated this protein the *P. xanthii* chitin-binding effector (PxCHBE).

### 3.3. PxCHBE Protein Is Deployed at the Plant Papilla Where Chitin Is Densely Accumulated

To gain further insight into the precise contribution of PxCHBE to *P. xanthii* pathogenesis, experiments were performed to determine the localization of PxCHBE. For comparison, the PxCDA1 protein was also included in these experiments. For this purpose, translational fusions of these proteins to CFP and GFP fluorescence markers were constructed, introduced into *P. xanthii* using the TGAT transformation method [11], and examined by CLSM (Figure 5). In the case of the PxCHBE-CFP fusion, CFP signals were observed under the fungal hyphae surrounding the penetration points in a location that resembled the location of the plant papilla (Figure 5B). In contrast, in the case of the PxCDA1-GFP fusion, GFP signals were observed to be associated with the fungal cell wall but only present in the cell wall of *P. xanthii* structures formed during the penetration of the plant cell, such as the appressorium (Figure 5A), penetration peg (Figure 5B), and immature haustorium (Figure 5C–E), and absent in the fully developed mature haustorium (Figure 5F).

To determine whether PxCHBE localization was indeed associated with plant papilla, Z-stack analysis of CFP signals was conducted and compared with those of the green fluorescence resulting from the binding of the WGA conjugate to free chitin in the plant papilla [27]. The analysis of the fluorescence patterns due to CFP and chitin were similar in morphology and distribution (Figure 6), confirming that the PxCHBE protein was located at the plant papilla at pathogen penetration sites and suggesting that the protein was secreted by the pathogen during the early stages of the interaction.

However, the expression analysis of *PxCHBE*, *PxCDA1*, and *PxCDA2* transcripts at different time points during the course of the infection process showed that the chitin-binding effector was expressed during the first stages of interaction in parallel with the expression of *PxCDAs* (Figure 7). However, *PxCHBE* expression was strongly repressed after reaching its maximum expression value at 48 h. In contrast, in the case of *PxCDAs*, their expression also decreased at 72 h but maintained certain levels of gene expression. These data were in agreement with the pattern of development of fungal structures observed by CLSM analysis.

### 3.4. CHBE Is a Novel Fungal Effector That May Be Present in Other Powdery Mildew Fungi

The structure of PxCHBE was computationally assessed by protein modelling. PxCHBE 3D models were constructed using the amino acid sequence without the signal peptide and several online analysis tools. Only AlphaFold2 and I-TASSER servers produced good quality models, with TM-score values of 0.70 and 0.73 and coverage values of 0.87 and 0.91, respectively. In both cases, the 3D models of PxCHBE were constructed using the crystal structure of a chitin deacetylase from *C. lindemuthianum* (2IW0) as a template. Furthermore, AlphaFold2 also predicted the formation of four disulfide bonds. As shown by the predicted models, PxCHBE and ECP6, a chitin-binding effector from *C. fulvum*, are proteins completely different in structure, with PxCHBE having no LysM domains involved in chitin binding (Appendix A).

Although PxCHBE does not present the typical LysM domains of other chitin-binding effector proteins, PxCHBE shows the amino acid residues involved in the binding to chitin in other proteins, such as lectins, which, like PxCHBE, also lack the LysM domain. However, despite sharing those amino acids, PxCHBE1 does not present any structural homology with those proteins. The putative chitin-binding site of PxCHBE was computationally assessed by molecular docking using the 3D models of PxCHBE predicted by AlphaFold and I-TASSER and the 3D models of the chitin pentamer, hexamer, and heptamer. Molecular docking analysis using the SwissDock server did not result in the identification of favourable binding sites. Since, as shown above, the PxCHBE protein expressed in vitro is capable of binding to chitin oligomers, the native fold of PxCHBE should be different from the native fold of the 3D models predicted for the protein. To elucidate the structure of PxCHEB, we tried to crystallize the protein, but it turned out to be toxic when expressed in yeast. In any case, although the structure of PxCHBE remains to be resolved, our data indicate that CHBE is a novel fungal chitin-binding effector and probably has a high affinity for chitin.

Finally, to determine the presence of proteins similar to CHBE in other fungi, BLAST searches were carried out using the sequence of the *PxCDA* gene as a query sequence to identify fungal genes with a similar intron distribution and the presence of a carbohydrate-binding module (CBM) in exon 1. Analysing the structure of the genes encoding CDA from different species of filamentous fungi and yeasts, a great variation was observed. We found that all of them have the chitin deacetylase domain. However, in the case of the CBM, important differences were found. Most have one or two CBMs, while rusts and yeasts do not, lacking the ability to produce a CHBE-like protein, as shown for *P. xanthii*. By analysing in detail the gene structure of the species presenting a CBM at the N-terminal domain, we found that only powdery mildew fungi such as *Blumeria graminis* and other species show the same gene structure as *P. xanthii CDA*, which suggests that these fungal species are likely to produce a CHBE protein.

## 4. Discussion

For the successful colonization of plant hosts by fungi, it is mandatory to suppress the activation of so-called chitin-triggered immunity. This strong defensive response is elicited after recognition by specific plant receptors of small chitin fragments, which are released by the activity of plant hydrolases such as chitinases, targeting the fungal cell wall during pathogen attack. To circumvent this immunity, phytopathogenic fungi have evolved different strategies aimed at avoiding chitin recognition or blocking signal transduction [43]. Despite the obvious need to suppress plant defences for the growth of obligate fungal biotrophs such as powdery mildew fungi, little is known about how these fungi overcome chitin-triggered immunity. The only exception is the cucurbit powdery mildew fungus *P. xanthii*. In this fungus, recent findings have shown the essential role of secreted proteins such as EWCA effectors and a haustorial-expressed LPMO enzyme in the breakdown of immunogenic chitin oligomers and the suppression of chitin signalling during pathogen penetration and haustorium development, respectively [27,28]. Similarly, in the same fungus, chitin deacetylation activity has also been shown to be critical during early stages of infection to circumvent chitin-triggered immunity [32].

In this work, we continued our analysis of the different counterstrategies employed by *P. xanthii* to subvert chitin-triggered immunity. We focused our attention on the role of a truncated *CDA* transcript. This transcript, originally designated *PxCDA3*, lacked most of the chitin deacetylase domain but retained the carbohydrate-binding module. Initially, considered a sequencing artefact, the transcript was consistently amplified from different cDNA samples, and expression analysis showed that it was highly expressed with an expression pattern typical of powdery mildew effectors [12]. The transcript appeared to be the result of alternative splicing of the *PxCDA* gene. Thus, the presence of an additional intron processing site in exon number 3 led to the loss of 22 nucleotides in this exon, resulting in a premature STOP codon in exon number 5 and producing a 156-amino-acid polypeptide, much smaller than typical fungal CDAs. Alternative splicing of chitin deacetylase genes appears to be a common phenomenon in chitin-containing organisms such as insects [44], and it is likely also to occur in fungi such as powdery mildews, as different *CDA* transcripts can be found in databases. A similar phenomenon seems to occur in nematodes as well. Interestingly, a major transcript was found in *Caenorhabditis elegans* that lacked the chitin deacetylase activity domain and had only the carbohydrate-binding domain [45], a finding similar to that reported here.

The selection of truncated versions of chitin deacetylase proteins suggests an adaptation phenomenon in powdery mildews to suppress chitin signalling. In a previous report, we showed that, unlike many other fungi, powdery mildews lack widely distributed LysM effectors [46] and secrete chitin-binding proteins that have a role in the sequestration of chitin oligosaccharides to dampen host defence. Here, we show that in the cucurbit powdery mildew *P. xanthii*, the shortest *CDA* transcript (*PxCDA3*) in fact encodes a secreted protein with chitin-binding capabilities similar to the chitin-binding capabilities of *C. fulvum* Ecp6 or *Magnaporthe oryzae* Slp1 proteins, that is, scavenging chitin polymers to prevent activation of chitin-triggered immunity [16,18,20,24]. This new protein, named CHBE (chitin-binding effector), has an amino acid sequence and a putative structure completely different from Ecp6, the most notable feature being the lack of LysM domains involved in chitin binding in Ecp6, indicating the different evolutionary origin of a protein with a similar activity. This phenomenon is known as convergent evolution: the same solution for a given problem but with proteins completely different in sequence and structure [47,48]. In other systems, such as the cacao pathogen *Moniliophthora perniciosa* and its sister species *M. roreri*, chitin-binding effectors have evolved as enzymatically inactive chitinases, a phenomenon known as the neofunctionalization of enzymes [25]. The independent evolution of different chitin-binding effectors suggests that these types of proteins have a critical role in fungal pathogenesis.

Localization studies using translational fusion proteins showed a pattern of distribution according to their biochemical function. Thus, while PxCDA1 was restricted to the fungal cell wall, PxCHBE was found in the plant papillae at pathogen penetration sites, where free chitin accumulates due to the activity of plant chitinases [27]. Furthermore, PxCDA1-GFP signals were observed from the first moments of penetration until the formation of a mature haustorium. However, PxCHBE-CFP fluorescence was observed only during papilla penetration, disappearing once the haustorium began to form. In addition, these data were supported by the expression patterns. Compared to *PxCDA1* and *PxCDA1* transcripts, *PxCHBE* expression was drastically reduced after 48 h when haustorium was assumed to be forming. In other words, *PxCHBE* showed a typical expression pattern of powdery mildew effectors that are induced in successive waves [12], the first wave being observed in this expression experiment. Moreover, the location of PxCHBE at plant papilla, a compartment that contains large amounts of plant hydrolases to prevent the establishment of infections [49,50], is in line with its biochemical features as a cysteine-rich protein with an even number of residues (10 cysteine residues). The presence of cysteine residues that can form disulfide bridges is a characteristic of secreted proteins, since these residues provide stabilization and protection against the activity of plant proteases [51]. Interestingly, AlphaFold2 predicts four disulfide bonds in the PxCHBE model protein. However, the lack of agreement between the experimental data and the in silico analysis means that the predicted model has to be different from the native fold. Therefore, the structure of PxCHBE is a pending issue and with it the role of disulfide bridges in protein stability.

One question that arose was whether PxCHBE and PxCDAs competed for similar substrates. However, their different locations suggest the use of different substrates. Scavenging of chitin to avoid plant recognition and the conversion of chitin to chitosan are necessary requirements for the development of the penetration peg and the formation of a mature haustorium. However, the action of plant chitinases is higher in the early stages of infection [52,53]. In a previous study, a peak in melon chitinase expression was observed in response to *P. xanthii* infection at 24 h postinoculation [28], which is precisely the moment of formation of the primary appressorium and penetration of the pathogen into the plant cell [54]. As shown here, this moment is when CHBE accumulates in plant papillae. This result is consistent with the result previously observed for EWCA effectors, chitinases secreted by *P. xanthii* to break down immunogenic chitin oligomers that also accumulate in plant papillae in the early stages of infection [27]. These results illustrate the presence of different *P. xanthii* effectors at precise times and places to disarm chitin signalling and suggest that effector deployment is perfectly orchestrated by host-imposed requirements. In this regard, Figure 8 shows the proposed role of the CHBE protein according to the results obtained in this work.

Our results have shown that *P. xanthii* contains a variety of proteins to neutralize chitin recognition by the host plant and that it does not relegate this function to a single protein. In previous work, we have shown that *P. xanthii* degradation of immunogenic oligomers and chitin deacetylation are key functions for survival and growth in the host [27,28,32]. In this work, we have shown that the sequestration of free chitin oligomers is also critical for the pathogenesis of *P. xanthii*. These results show the importance of the pathogen going unnoticed by the host, especially in the early stages of infection. Previous work on *C. fulvum* has shown that in addition to Ecp6, the pathogen presents another effector with the ability to bind to chitin, Avr4. This effector has affinity for chitin, such as Ecp6 or CHBE, but unlike these, it does not sequester free chitin oligomers but binds to chitin in the fungal cell wall, protecting it from the activity of plant chitinases [21,55]. The fact that *P. xanthii* and other fungi possess different proteins destined for the same function, preventing chitin recognition, is not surprising since in nature, functions important for survival are often redundant [56,57]. Finally, BLAST searches showed that CHBE is likely to occur only in powdery mildew species. However, it is tempting to speculate that other fungi, especially those that do not have LysM effectors, likely employ analogous proteins. As previously suggested for neofunctionalization [25], alternative splicing may be an evolutionary pathway for the rise of new virulence factors in fungi.

## Figures and Tables

**Figure 1 jof-08-01022-f001:**
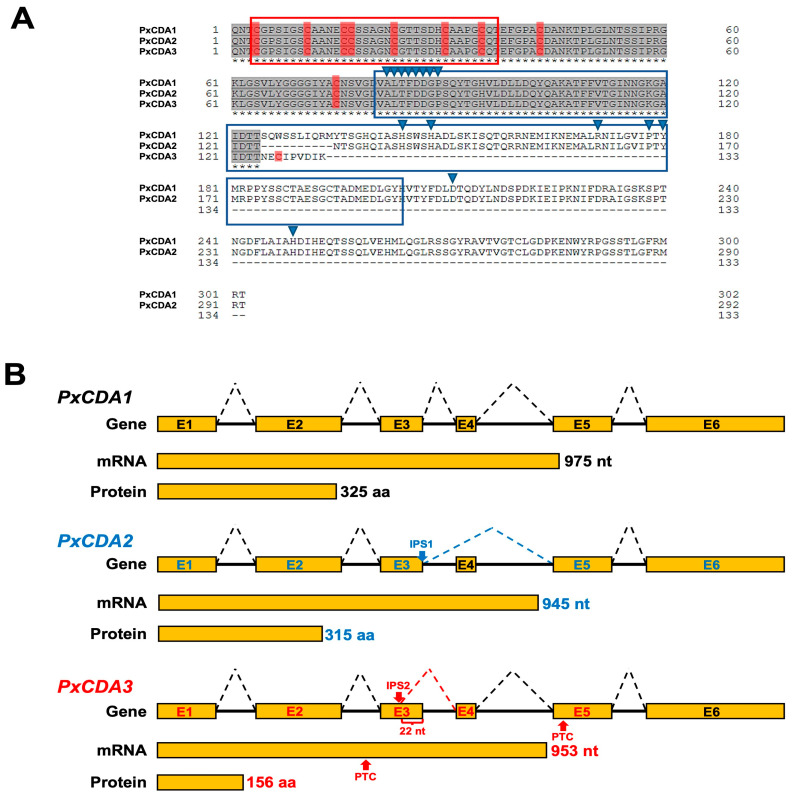
Alternative splicing of the *PxCDA* gene generates a CDA variant and a putative chitin-binding protein. (**A**) Amino acid sequence alignment of the mature proteins deduced from the three *P. xanthii* transcripts encoding chitin deacetylase sequences. Identical amino acids are shown in grey. The carbohydrate-binding module (CBM) and chitin deacetylase (CDA) domain are depicted in red and blue boxes, respectively. The blue arrowheads show the residues important for CDA activity. The cysteine residues are shown in red. The sequences were aligned using the UniProt web server. (**B**) Schematic representation of the alternative splicing events that generate alternative splicing variants from the *PxCDA* gene. Note how the PxCDA3 protein seems to be an alternative splicing variant with features of a cysteine-rich protein that retains the CBM module and lacks a substantial part of the chitin deacetylase domain. Abbreviations: E, exon; nt, nucleotide; aa, amino acid; IPS, intron processing site; PTC, premature termination codon.

**Figure 2 jof-08-01022-f002:**
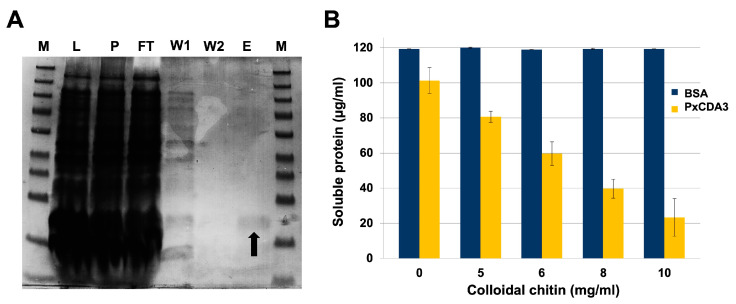
Chitin-binding activity of His-tagged PxCDA3. (**A**) In vitro expression and purification of His-tagged PxCDA3 protein. Images show protein samples at different purification stages separated by SDS-PAGE. The arrow denotes the band corresponding to the purified His-tagged protein PxCDA3. The lanes are as follows: P, pellet sample; L, supernatant sample of the cell lysate; FT, discarded flow-through sample after protein binding to the affinity column; W1, first washing sample; W2, second washing sample; E, eluted protein. M, Spectra multicolour broad range protein ladder (Thermo-Fisher). (**B**) Chitin-binding activity of His-tagged PxCDA3 protein. Quantification of soluble proteins (supernatant fraction) after incubation of soluble His-tagged PxCDA3 with different concentrations of colloidal chitin for 60 min. BSA was used as a negative control. Bars indicate the standard error of three technical replicates from three different experiments.

**Figure 3 jof-08-01022-f003:**
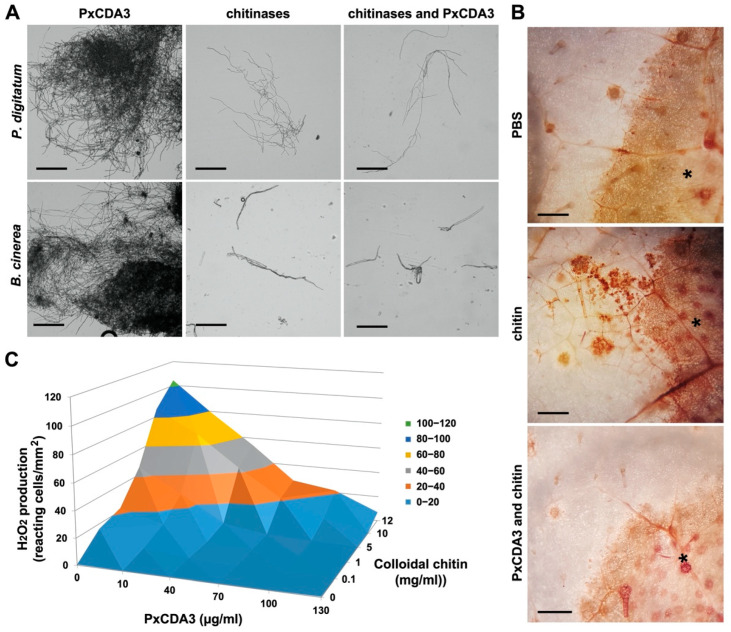
His-tagged PxCDA3 does not bind chitin in fungal cell walls but prevents chitin recognition by the plant. (**A**) Assay of chitinase activity against fungal cell walls. The filamentous fungal pathogens *P. digitatum* and *B. cinerea* were grown in PDB and CZAPEK broth, respectively, and incubated with His-tagged PxCDA3 and *S. griseus* chitinases. Micrographs were taken 24 h after incubation. Representative micrographs from three independent experiments are shown. Scale bars: 200 μm. (**B**) Activation of chitin-triggered immunity (oxidative burst) and suppression of this response by His-tagged PxCDA3. Melon leaves were infiltrated with colloidal chitin and His-tagged PxCDA3. Detection of H_2_O_2_ was performed by the DAB uptake method. As positive and negative controls for the production of oxidative bursts in leaf tissue, colloidal chitin and PBS were used, respectively. Photographs were taken 48 h after infiltration. Asterisks denote the infiltration sites. Epidermal cells with reddish-brown precipitates are reactive cells showing accumulation of H_2_O_2_. Scale bars: 250 μm. (**C**) Hydrogen peroxide accumulation by epidermal cells upon infiltration of different concentrations of colloidal chitin in combination with different concentrations of His-tagged PxCDA3. The data were recorded 48 h after infiltration. Data represent the mean of three independent assays.

**Figure 4 jof-08-01022-f004:**
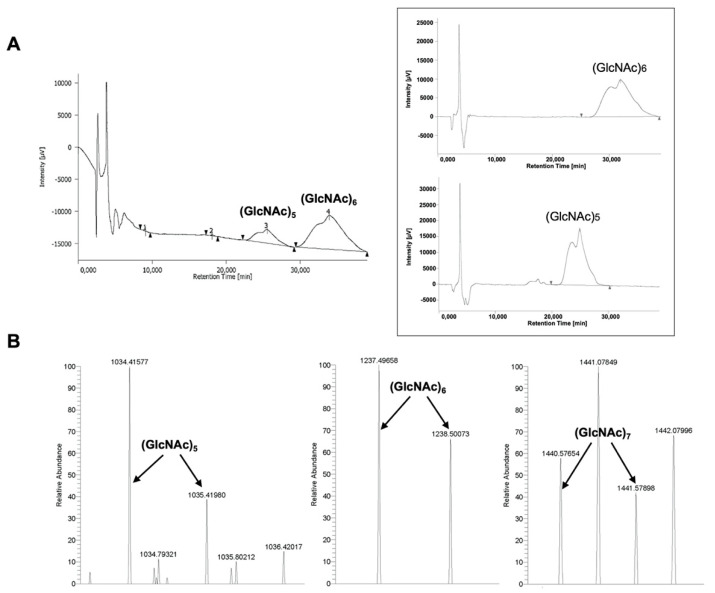
Chitin-binding activity of His-tagged PxCDA3 on different chitin oligomers. For these assays, His-tagged proteins were maintained bound to nickel affinity columns and incubated with a mix of chitin oligomers (GlcNAc)_1–7_. After washing the columns and elution and denaturation of the proteins, the content of oligomers retained by the proteins was analysed. (**A**) Analysis of chitin oligomers by HPLC. Chromatograms corresponding to reference oligomers are shown in a box on the right. A representative chromatogram from three independent experiments is shown. The peaks corresponding to chitin pentamers and hexamers are indicated. (**B**) Analysis of chitin oligomers by H-ESI-MS. Representative chromatograms from three independent experiments are shown. The peaks corresponding to chitin pentamers, hexamers, and heptamers are indicated.

**Figure 5 jof-08-01022-f005:**
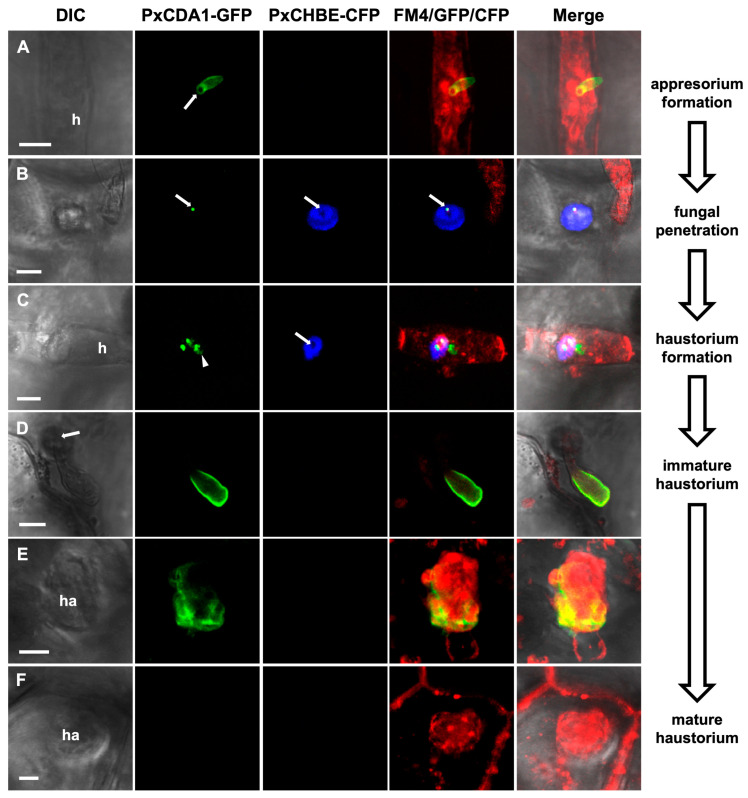
Localization analysis of PxCHBE-CFP and PxCDA1-GFP fusions. *Podosphaera xanthii* colonies growing in melon plants were co-transformed with the translational fusions contained in the T-DNA regions of pPxCHBE-CFP and pPxCDA1-GFP using the TGAT system. Transformed colonies were stained with the membrane-specific stain FM4-64 (red) and examined by CLSM. The images provided show different stages of fungal development, including appressorium formation (**A**), fungal penetration (**B**), and haustorium formation (**C**–**F**), from which differential interference contrast (DIC), fluorescence CLSM, and merged images are shown. While PxCDA1-GFP signals (green) are observed in different fungal development structures, PxCHBE-CFP fluorescence (blue) is restricted to fungal penetration and the early stages of haustorium formation. Arrows denote penetration points. Scale bars: 5 µm. Abbreviations: h, hypha; ha, haustorium.

**Figure 6 jof-08-01022-f006:**
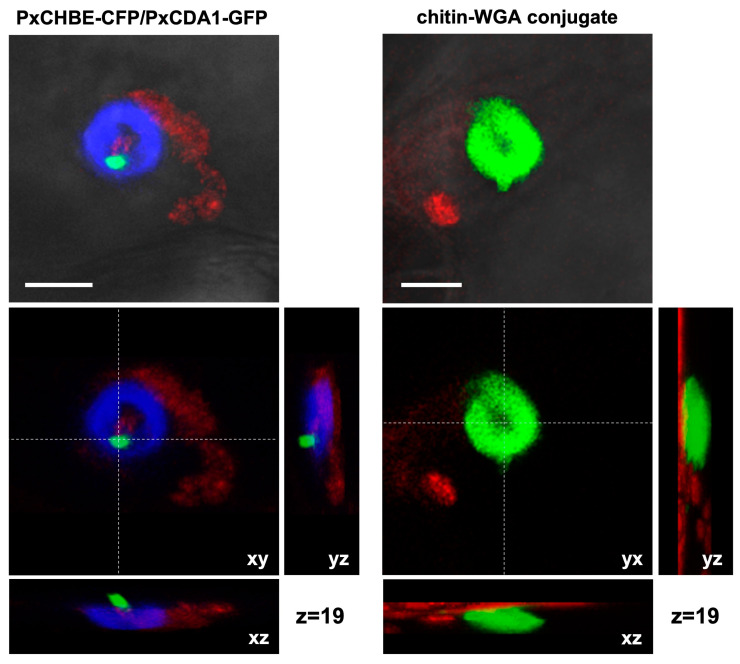
Detailed localization analysis of PxCHBE-CFP and PxCDA1-GFP fusions and free chitin released in plant papilla. *Podosphaera xanthii* colonies growing in melon plants were co-transformed with the T-DNA regions of pPxCHBE-CFP and pPxCDA1-GFP using the TGAT system. Free chitin was stained using WGA-Alexa Fluor 488 conjugate (green). Transformed colonies were stained with the membrane-specific stain FM4-64 (red) and examined by CLSM. Top images: detailed views of the PxCHBE-CFP (blue) and PxCDA1-GFP (green) signals and green fluorescence of chitin-WGA conjugate localized under the hypha. The PxCHBE-CFP fluorescence area shows a hole in the centre corresponding to the penetration site where a spot of the PxCDA-GFP signal was observed. A similar distribution pattern was observed for the green fluorescence signal attributable to free chitin in the plant papilla. Bottom images: CLSM z-stack micrographs of the PxCHBE-CFP/PxCDA1-GFP and chitin signals showing orthogonal views from different planes (xy, xz, and yz). The similar location of PxCHBE-CFP and chitin signals indicates that PxCHBE is deployed in the plant papilla. Scale bar: 5 µm.

**Figure 7 jof-08-01022-f007:**
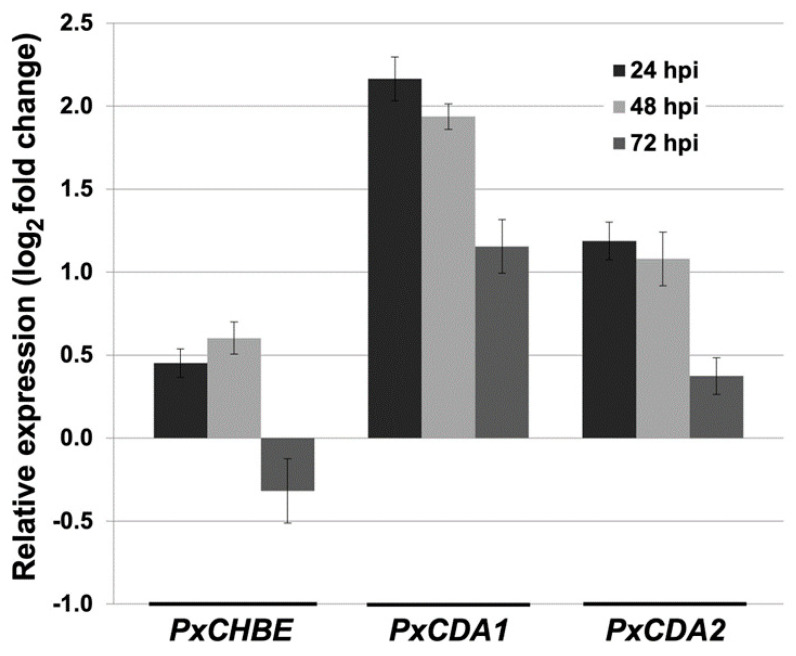
Time-course analysis of the expression of the *PxCHBE*, *PxCDA1*, and *PxCDA2* transcripts. Total RNA was isolated from melon cotyledons infected with *P. xanthii* at different time points, and the relative expression of the *PxCHBE*, *PxCDA1*, and *PxCDA2* transcripts was analysed by quantitative RT-PCR. Transcript abundance was normalized to the abundance of the endogenous control β-tubulin gene *PxTUB2* (KC333362). The relative expression of each gene was calibrated to the sample 0 h postinoculation. The data shown represent the average values of four technical replicates from three independent experiments, with error bars showing the standard error.

**Figure 8 jof-08-01022-f008:**
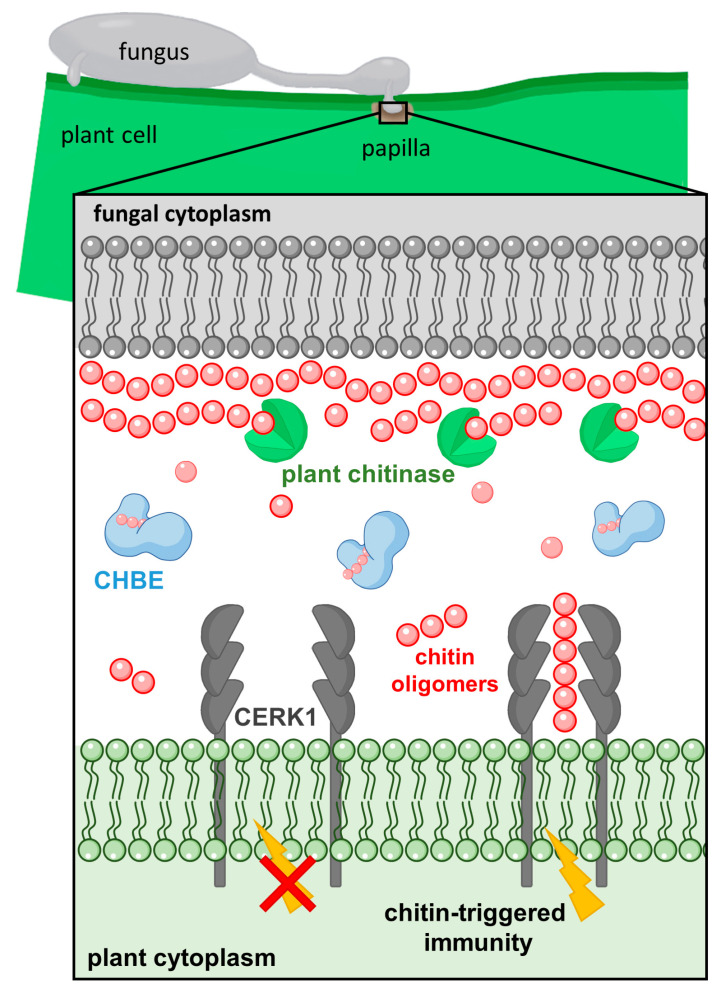
Schematic representation of the proposed role for CHBE proteins. At the papilla (fungal pathogen penetration site), plant-secreted endochitinases release chitin fragments from fungal cell walls that can be recognized by the plant CERK1, activating chitin-triggered immunity. To counteract this action, CHBE is released by the fungus at the same sites to sequester immunogenic chitin oligomers, which cannot induce dimerization of CERK1, thus suppressing the activation of chitin signalling. To reduce complexity, many components of the fungal and plant cell walls have been omitted.

**Table 1 jof-08-01022-t001:** Chitin-binding activity of His-tagged PxCDA3 on different chitin oligomers.

Assay	Substrate ^a^	Products (µg mL^−1^) ^b^						
		Mono	Di	Tri	Tetra	Penta	Hexa	Hepta
Experiment 1	(GlcNAc)_1–7_	- ^c^	-	-	-	57	235	Nd ^d^
Experiment 2	(GlcNAc)_1–7_	-	-	-	-	50	186	nd
Experiment 3	(GlcNAc)_1–7_	-	-	-	-	49	193	nd

^a^ A mixture of chitin (GlcNAc) oligomers (mono, di, tri, tetra, penta, hexa, and hepta) was assayed at a final concentration of 500 µg mL^−1^. ^b^ Concentrations of different GlcNac oligomers at the end of the assay were determined by HPLC analysis. ^c^ Concentrations below detection levels (0.1 µg mL^−1^). ^d^ Non detected by HPLC analysis.

## Data Availability

Not applicable.

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
