# Peer review of "Suppression of Chitin-Triggered Immunity by a New Fungal Chitin-Binding Effector Resulting from Alternative Splicing of a Chitin Deacetylase Gene"

_jof, 2022, doi:10.3390/jof8101022_

Round 1
Reviewer 1 Report
Assessment of the manuscript jof-1924500, entitled Suppression of Chitin-Triggered Immunity by a New Fungal Chitin-Binding Effector Resulting from Alternative Splicing of a Chitin Deacetylase Gene
I have to say, an elaborate, interesting and relevant manuscript which is in my view suitable for publication in JoF. However, I do have some issues.
The first issues concern Figure 4 and the related Results part. In 4A the peaks of the pentamers and hexamers are indicated. These peaks are wide. How did the authors determine that these were the actual pentamers and hexamers. Is that merely based on 4B (Mass spec) or were the oligomers used in the HPLC analysis used as well as a reference? Was an array detector used to for the spectral pattern and, if so, could these oligomers be distinguished by their individual spectral pattern (The authors only used a mix of mono- and oligomers, right?)? In the latter case, is including Figure 4A really that relevant? From the materials and methods it is however not clear The chromatogram indicating the hexamers is very clean but those indicating the penta- and heptamers show some background. Particularly the chromatogram with the indicated heptamers, show two additional significant peaks. It therefore seems that PxCBE also has an affinity for those molecules as well. Has it been determined what those are?
The second issue concerns Figure 8. The authors used two protein folding algorithms (8A) and the show how different this folding is when compared to CfECP6 which was acquired from the Protein Data Bank. However, there is no indication what folding algorithm was used for CfECP6, nor was CfECP6 folded by the authors themselves using the algorithms used for PxCHEB. Added to this, the position of the Cysteines or the location of the disulfide bridges (as anticipated, there must be 5 of them) have not been indicated as well. Thus, how solid (and relevant) are these models?
And then there is the question what may be the reason why PxCDA1 and -2 have only 9 Cysteine residues. These proteins may be larger but are also extracellular, as PxCHEB is. This is rather scarcely discussed.
The English can be improved here and there (see also Specific comments).
Specific comments:
Introduction part.
- Third line: "grapevine" instead of "grapevines"
- “Biotrophy in these fungi is the result of developing haustoria inside plant cells, specialized structures of parasitism that are responsible for nutrient uptake and factorexchange with the plant [3,4].” Not a nice sentence, particularly the word “factor”. To vague, rephrase.
- “……arguably the group more severely affected.” “Better is ”…..arguably the group most affected”.
- “….., our laboratory has invested remarkable efforts in producing fundamental research resources…..” strange sentence, particularly the word “remarkable”. All not that relevant. Rephrase.
- “referred to as microbe- or pathogen-associated molecular patterns (PAMPs).” Better would be “referred to as microbe- or pathogen-associated molecular patterns (MAMPs or PAMPs).”
- “To help this aspect, the fungal effectors, small proteins secreted by fungal pathogens, play a key role,……” Strange (start of the ) sentence. Rephrase.
Results part.
- “…..,so they do not have the ability to produce a CHBE-like protein,.….” Rephrase, without using “so”.
Author Response
DEAR EDITOR AND REVIEWERS:
We felt that your comments were very appropriate and helpful to improve the quality of our manuscript. Next, we explained, point-by-point, our responses (red color) to your interesting comments, which are also highlights in yellow and green colors through the manuscript:
REFEREE 1
The first issues concern Figure 4 and the related Results part. In 4A the peaks of the pentamers and hexamers are indicated.
Q1. These peaks are wide. How did the authors determine that these were the actual pentamers and hexamers. Is that merely based on 4B (Mass spec) or were the oligomers used in the HPLC analysis used as well as a reference?
The oligomers were used as well as a reference in the HPLC analysis. Chromatograms obtained for reference samples corresponding to pentamers and hexamers have been included in Figure 4A in a box on the right. Compared to reference samples, the peaks are not that wide.
Q2. Was an array detector used to for the spectral pattern and, if so, could these oligomers be distinguished by their individual spectral pattern (The authors only used a mix of mono- and oligomers, right?)?
Yes. As shown in Figure 4A, they can be clearly distinguished.
Q3. Is including Figure 4A really that relevant?
The Figure 4A is an illustration of Table 1 data. It is relevant for us; it helps to understand Table 1.
Q4. The chromatogram indicating the hexamers is very clean but those indicating the penta- and heptamers show some background. Particularly the chromatogram with the indicated heptamers, show two additional significant peaks. It therefore seems that PxCBE also has an affinity for those molecules as well. Has it been determined what those are?
Those peaks were not checked. Indeed, by both analyses, we concluded that PxCHBE has affinity at least for penta, hexa and hepta chitin oligomers.
The second issue concerns Figure 8.
Q5. The authors used two protein folding algorithms (8A) and the show how different this folding is when compared to CfECP6 which was acquired from the Protein Data Bank. However, there is no indication what folding algorithm was used for CfECP6, nor was CfECP6 folded by the authors themselves using the algorithms used for PxCHEB.
As indicated in the figure legend, the structure of the chitin-binding effector ECP6 from C. fulvum (CfECP6, 4B9H) was obtained from the Protein Data Bank. That means it is based on the protein crystal and not on a folding algorithm.
Q6. Added to this, the position of the Cysteines or the location of the disulfide bridges (as anticipated, there must be 5 of them) have not been indicated as well. Thus, how solid (and relevant) are these models?
As suggested, disulfide bonds have now been indicated in the protein models figure, but only in the AlphaFold2 model of PxCHBE because I-TASSER does not allow the visualization of disulphide bonds. As shown in Figure S2B, four disulphide bonds are expected to be formed. However, as indicated in the text, the in silico analysis (protein modelling and molecular docking) is not consistent with experimental data, indicating that the native fold of PxCHBE should differ from the predicted models. We thought of this figure to show that PxCHBE is completely different chitin-binding protein from typical LysM effectors. Since these protein models are indeed not solid, we propose move this figure to supplementary material as the Figure S2.
Q7. And then there is the question what may be the reason why PxCDA1 and -2 have only 9 Cysteine residues. These proteins may be larger but are also extracellular, as PxCHEB is. This is rather scarcely discussed.
Indeed PxCDA1 and PxCDA2 are extracellular proteins, but not all extracellular proteins are cysteine-rich proteins. These proteins, at least PxCDA1, are not secreted at the papilla; they are mostly associated with haustorium cell wall.
Q8. The English can be improved here and there (see also Specific comments).
American Journal Experts edited the manuscript. Editing suggestions have been all included.
Specific comments:
All changes were made and highlights through the manuscript in yellow color.
Introduction part.
- Third line: "grapevine" instead of "grapevines"
- “Biotrophy in these fungi is the result of developing haustoria inside plant cells, specialized structures of parasitism that are responsible for nutrient uptake and factor exchange with the plant [3,4].” Not a nice sentence, particularly the word “factor”. To vague, rephrase.
- “……arguably the group more severely affected.” “Better is ”…..arguably the group most affected”.
- “….., our laboratory has invested remarkable efforts in producing fundamental research resources…..” strange sentence, particularly the word “remarkable”. All not that relevant. Rephrase.
- “referred to as microbe- or pathogen-associated molecular patterns (PAMPs).” Better would be “referred to as microbe- or pathogen-associated molecular patterns (MAMPs or PAMPs).”
- “To help this aspect, the fungal effectors, small proteins secreted by fungal pathogens, play a key role,……” Strange (start of the ) sentence. Rephrase.
Results part.
- “…..,so they do not have the ability to produce a CHBE-like protein,.….” Rephrase, without using “so”.

Reviewer 2 Report
Dear Authors,
I appreciate your comprehensive work and writing in the manuscript which informed a New Fungal Chitin-Binding Effector and how the alternative splicing of a Chitin Deacetylase Gene involved on it.
I put some comment to improve your manuscript. Please find on the page 4,10 and 13.
I agree that this manuscript is accepted to be published, after the minor revision.

Author Response
DEAR EDITOR AND REVIEWERS:
We felt that your comments were very appropriate and helpful to improve the quality of our manuscript. Next, we explained, point-by-point, our responses (red color) to your interesting comments, which are also highlights in yellow and green colors through the manuscript:
REFEREE 2.
Q1 (page 4). Details on inoculation of the pathogen are provided:
“For the inoculation, P. xanthii conidia were collected by immersing infected zucchini cotyledons in 50 mL of a 0.01% Tween-20/distilled water solution. Melon cotyledons were spray-inoculated with a spore suspension at 1 × 106 conidia mL−1. Plants were then incubated in a growth chamber under the conditions mentioned above.”
Q2 (page 12). A sentence has been added according to suggestion.
“Epidermal cells with reddish-brown precipitates are reactive cells showing accumulation of H2O2.”
Q3 (page 12). Hydrogen peroxide is one of the reactive oxygen species (ROS). To avoid confusion, we have eliminated the term “ROS” from the legend of Figure 3C and from the plot of Figure 3C. We also indicate in the text (page 11) that accumulation of H2O2 is an example of ROS.
Q4 (page 13). This suggestion should be included in the Discussion section and not in the Results. To explain that we have included a sentence in page 17:
“In other words, PxCHBE showed a typical expression pattern of powdery mildew effectors that are induced in successive waves [12], the first wave being observed in this expression experiment.”
